# Prognostic Value of the Lactate/Albumin Ratio in Sepsis-Related Mortality: An Exploratory Study in a Tertiary Care Center with Limited Resources in Western Mexico

**DOI:** 10.3390/jcm14082825

**Published:** 2025-04-19

**Authors:** Octavio Ponce-Orozco, Berenice Vicente-Hernandez, Sol Ramirez-Ochoa, Luis Asdrúval Zepeda-Gutiérrez, Mauricio Alfredo Ambriz-Alarcón, Gabino Cervantes-Guevara, Alejandro Gonzalez-Ojeda, Clotilde Fuentes-Orozco, Francisco Javier Hernandez-Mora, Guillermo A. Cervantes-Cardona, Janet Cristina Vazquez-Beltran, Enrique Cervantes-Pérez

**Affiliations:** 1Department of Internal Medicine, Hospital Civil de Guadalajara Fray Antonio Alcalde, Centro Universitario de Ciencias de la Salud, Universidad de Guadalajara, Guadalajara 44280, Mexico; octaponce.20@gmail.com (O.P.-O.); dra.berenicevicente@gmail.com (B.V.-H.); sramirez@hcg.gob.mx (S.R.-O.); luis-zent@hotmail.com (L.A.Z.-G.); 2Subdirección Médica, Hospital Civil de Guadalajara Fray Antonio Alcalde, Guadalajara 44280, Mexico; mau_ambriz@hotmail.com; 3Department of Gastroenterology, Hospital Civil de Guadalajara Fray Antonio Alcalde, Guadalajara 44280, Mexico; gabino_guevara@hotmail.com; 4Department of Welfare and Sustainable Development, Centro Universitario del Norte, Universidad de Guadalajara, Colotlan 46200, Mexico; 5Biomedical Research Unit 02, Hospital de Especialidades, Centro Médico Nacional de Occidente, Guadalajara 44350, Mexico; avygail5@gmail.com (A.G.-O.); clotilde.fuentes@gmail.com (C.F.-O.); 6Department of Human Reproduction, Growth and Child Development, Health Sciences University Center, Universidad de Guadalajara, Guadalajara 44280, Mexico; frank.gine@gmail.com; 7Department of Philosophical, Methodological and Instrumental Disciplines, Centro Universitario de Ciencias de la Salud, Universidad de Guadalajara, 44340, Mexico; gacervantes66@hotmail.com; 8School of Medicine, Instituto Politecnico Nacional, Mexico City 11340, Mexico; janet.cris.beltran@gmail.com; 9Departamento de Clínicas, Centro Universitario de Tlajomulco, Universidad de Guadalajara, Tlajomulco de Zuniga 45641, Mexico

**Keywords:** lactate, sepsis, albumin

## Abstract

**Background and Objectives**: Sepsis is a critical condition with high mortality rates worldwide. The early identification of patients at an elevated risk of mortality remains a significant clinical challenge. The lactate/albumin (L/A) ratio has emerged as a potential prognostic biomarker in critically ill patients. This exploratory study aimed to evaluate the L/A ratio as a predictor of mortality in patients with sepsis or septic shock and to describe the demographic and clinical characteristics of affected patients in a tertiary referral hospital in Western Mexico. **Materials and Methods**: A retrospective cross-sectional study was conducted including patients diagnosed with sepsis or septic shock between January 2022 and June 2023. Clinical and biochemical data, including serum lactate and albumin levels, were collected from medical records to calculate the L/A ratio. The primary outcome was in-hospital mortality. The statistical analysis included receiver operating characteristic (ROC) curves to evaluate the L/A ratio’s discriminative capacity, a bivariate analysis, and a multivariate logistic regression to identify independent risk factors for mortality. **Results**: A total of 141 patients were included in the study, and the median L/A ratio was significantly higher in non-survivors compared to survivors (0.95 vs. 0.60, *p* = 0.003). The ROC analysis showed an area under the curve (AUC) of 0.651, with a sensitivity of 36.5% and specificity of 82% at a cutoff value of 1.12. The multivariate analysis identified serum albumin levels and vasopressor use as independent predictors of mortality. **Conclusions**: The L/A ratio demonstrates potential as a simple and accessible prognostic biomarker for mortality in sepsis, though its utility requires further validation in larger prospective studies.

## 1. Introduction

Sepsis is a dysregulated host response to infection that causes life-threatening organ failure [1]. Septic shock is a form of sepsis with circulatory, metabolic, and cellular dysfunction and higher mortality. In 2017, the Global Burden of Disease Study reported 11 million sepsis-related deaths and 48.9 million new cases [2]. Gorordo-Delsol et al. recently conducted Mexico’s largest sepsis study [3]. In this cross-sectional research of 3,279 patients in 68 emergency departments, 307 had sepsis (12.9%), and the reported death rate was 16.93%: 9.39% for sepsis and 65.85% for septic shock.

The early detection and treatment of sepsis within the first few hours improve patient outcomes [4]; however, clinically identifying high-risk patients remains difficult [5]. Multiple clinical scores and biomarkers have been used to identify and stratify sepsis patients [5]. Cost-effective biomarkers with high sensitivity and specificity for high-risk individuals are being studied [4,5,6]. Although several biomarkers, such as procalcitonin and C-reactive protein, among others, have shown usefulness in diagnosing, prognosticating, and guiding antimicrobial treatment in sepsis, their individual effectiveness varies widely between studies [5].

Early warning scores (EWS) are scoring systems that rapidly and quantitatively measure physiologic and vital sign changes that may indicate clinical deterioration [7]. The National Early Warning Score (NEWS) [8], NEWS2 [9], Modified Early Warning Score (MEWS) [10], Rapid Acute Physiology Score (RAPS) [11], Rapid Emergency Medicine Score (REMS) [12], and Simple Early Warning Score (SEWS) [13] are the most predictive and commonly used of these scores. Additionally, the Sequential Organ Failure Assessment (SOFA) score [14], the Acute Physiology and Chronic Health Evaluation II (APACHE II) [15], and the Simplified Acute Physiology Score 3 (SAPS 3) [16] have been used to predict outcomes in critically ill patients. Another commonly used mortality predictor in critically ill patients is the quick SOFA (qSOFA) score—a simplified version of SOFA that has been extensively validated [17,18,19]. However, one of its major limitations is that it solely relies on clinical variables, making its reliability dependent on the accuracy of the physician or nurse recording these signs.

The complexity and quantity of factors required to measure these clinical scores make their effective implementation difficult in all medical contexts, especially in high-patient-concentration facilities with limited personnel and resources. Besides clinical scores, biomarkers including mid-regional pro-adrenomedullin (MR-proADM) [20], albumin [5,21], and serum lactate levels [21,22] have proven useful as independent prognostic indicators; however, they may lack the specificity needed for widespread usage, and measuring some of them at hospitals with limited clinical laboratories may be difficult. This is widespread in low- and middle-income countries with limited healthcare resources, such as Mexico [23].

Elevated serum lactate levels alone are substantially connected with mortality and used for early sepsis and septic shock diagnosis, therapy, and risk stratification [6,22,24,25,26]. However, liver and kidney dysfunction, heart injury, heart failure, acute pulmonary edema, increased glycolysis, medication use (e.g., metformin, albuterol, and epinephrine), and metabolic alterations, such as those caused by malignant neoplasms and diabetic ketoacidosis, can cause hyperlactatemia [22,26]. In sepsis, poor peripheral oxygenation causes anaerobic glycolysis and lactate buildup [27,28,29]. Due to confounding factors that affect lactate concentrations [6], its predictive value is currently debated [30].

Besides metabolic changes caused by hypoperfusion, the elevation of systemic inflammation markers in patients with sepsis, such as cytokines, leads to damage to the vascular endothelium, increased capillary permeability, renal dysfunction, and lower serum albumin levels, often resulting in hypoalbuminemia [31]. Chronic disorders often are accompanied by hypoalbuminemia, which, in sepsis, has been associated with a poor prognosis [27,31,32,33,34]. A prospective research study by Yin et al. [30] found that albumin levels below 2.9 mg/dL (95% CI: 0.620–0.829; *p* < 0.001) significantly predicted the 28-day mortality of sepsis patients, with a sensitivity of 0.718 and specificity of 0.677.

Research into the simultaneous use of lactate and albumin in search of improving their performance as mortality predictors has been the target of several studies [27,31,33,34]. Current research suggests that the lactate/albumin (L/A) ratio may predict multiple organ failure and mortality in critically ill patients, including patients with sepsis [35].

The usefulness of the L/A ratio as a predictor of mortality in sepsis has been investigated in pediatric populations [27] and in adult populations [35,36], with generally favorable results and demonstrating better performance compared to lactate or albumin levels individually [36,37,38,39,40,41,42,43,44]. Reported mean L/A values in sepsis patients range from 0.64 to 1.8, while among non-survivors and those with multiple organ failure, the reported range increases to 0.85 to 2.29 [27,34,36,38,45,46].

While these findings suggest that the L/A ratio is a promising prognostic tool, there remains a lack of extensive prospective research to validate its clinical application. Additionally, data on its prognostic utility in Latin American populations remain scarce; to our knowledge, there is only information from three other studies regarding the prognostic utility of the L/A ratio in Mexican patients with sepsis [47,48,49].

The objective of this exploratory study was to evaluate the usefulness of the L/A ratio as a predictor of mortality in patients with a diagnosis of sepsis admitted to a hospital unit in Western Mexico, providing valuable epidemiological insights into sepsis outcomes in resource-limited settings [23].

## 2. Materials and Methods

### 2.1. Subjects and Study Design

This retrospective cross-sectional study included patients diagnosed with sepsis, including those who developed septic shock, based on the criteria established by the Surviving Sepsis Campaign 2021 [6]. Patients were recruited from the Internal Medicine Service at Hospital Civil de Guadalajara Fray Antonio Alcalde between January 2022 and June 2023. Eligible patients were those aged 18 years or older, of any sex, whose serum lactate and albumin levels were measured upon hospital admission. Patients with a previous diagnosis of chronic kidney disease, nephrotic syndrome, liver failure, or chronic malnutrition were excluded from the study, as these conditions are strongly associated with hypoalbuminemia [43,46]. Although patients with conditions known to independently elevate serum lactate levels were not actively excluded, none of the individuals included in the study had a documented diagnosis of heart failure, myocardial injury, pulmonary edema, malignancy, thiamine deficiency, or mitochondrial disease [50].

### 2.2. Clinical and Biochemical Variables Analyzed

Clinical and biochemical data were collected from electronic medical records. The clinical variables included age, sex, smoking status, weight, height, body mass index (BMI), history of diabetes mellitus, arterial hypertension, stroke, and other comorbidities. In addition, sepsis-related variables such as primary site of infection, use of mechanical ventilation, and vasopressor requirements during hospitalization were recorded. Laboratory parameters, including serum lactate and albumin levels, were obtained from routine blood tests performed at hospital admission, and these values were used to calculate the lactate/albumin (L/A) ratio. All biochemical analyses were conducted following standard methodologies in the hospital’s clinical laboratory.

The qSOFA score was calculated as described by Seymour CW et al. [17]. This score evaluates three clinical parameters: altered mental status (Glasgow Coma Scale < 15), respiratory rate ≥ 22 breaths per minute, and systolic blood pressure ≤ 100 mmHg. Each parameter meeting the criteria was assigned one point, and patients with a qSOFA score of 2–3 were classified as high risk for in-hospital mortality.

### 2.3. Data Collection and Statistical Analysis

All data were entered into a database using Microsoft Office Excel 2016 (Microsoft Corporation, Redmon, WA, USA), and statistical analyses were performed using SPSS Statistics (version 26, IBM, Armonk, NY, USA). Descriptive statistics were applied to summarize the data, with absolute and relative frequencies (percentages and proportions) used for qualitative variables, and medians with interquartile ranges (IQRs) applied to quantitative variables. For inferential statistics, categorical variables were analyzed using chi-square tests in 2 × 2 contingency tables. The normality of continuous variables was assessed using D’Agostino–Pearson and Kolmogorov–Smirnov tests. Given the non-normal distribution of most quantitative variables, Mann–Whitney U tests were used to compare unpaired groups, with the median as the comparison metric.

To identify factors associated with mortality, a multivariate analysis was performed using binary logistic regression with a backward stepwise model (likelihood ratio method). The Hosmer–Lemeshow goodness-of-fit test was applied to assess the model’s adequacy. All predictor factors that were statistically significant in the bivariate analysis were included in the model, and the odds ratios (ORs) with 95% confidence intervals (CIs) were computed to measure the strength of associations.

To evaluate the prognostic performance of the L/A ratio, a receiver operating characteristic (ROC) curve analysis was conducted. The area under the curve (AUC) was calculated to assess its discriminative ability in predicting mortality. The optimal cutoff value for the L/A ratio was determined based on the sensitivity and specificity in distinguishing survivors from non-survivors using Youde’s J statistic. A *p*-value < 0.05 was considered statistically significant for all analyses.

### 2.4. Ethical Considerations

This study was conducted in accordance with the Declaration of Helsinki and Good Clinical Practice (GCP) guidelines. The study protocol was reviewed and approved by the Clinical Research and Bioethics Committee of Hospital Civil de Guadalajara Fray Antonio Alcalde (approval code: CEI-27/24). Given the retrospective nature of the study, the requirement for informed consent was waived by the ethics committee.

## 3. Results

A total of 141 patients diagnosed with sepsis who met the inclusion criteria and did not meet any exclusion criteria were included in the study. The clinical history and demographic characteristics of the patients are presented in Table 1. Among these patients, 56.7% were men and 43.3% were women. Patients were categorized into survivors (*n* = 89; 52.8% men, median age = 48 years, IQR = 38.5, 60) and non-survivors (*n* = 52; 63.5% men, median age = 53 years, IQR = 42.25, 60.75).

The differences in clinical and laboratory characteristics between survivors and non-survivors are presented in Table 2. Of the total 141 patients included in the study, 21.9% required mechanical ventilation during their hospital stay.

To evaluate the prognostic accuracy of the L/A ratio as a mortality predictor, a receiver operating characteristic (ROC) curve was generated, and the area under the curve (AUC) was calculated. The AUC for the L/A ratio in patients with sepsis was 0.651 (95% CI: 0.55–0.74). The optimal cutoff value of the L/A ratio to differentiate survivors from non-survivors was 1.12, yielding a sensitivity of 36.5%, specificity of 82%, positive predictive value (PPV) of 39%, and negative predictive value (NPV) of 84%. A graphical representation of the ROC curve is shown in Figure 1.

To identify independent factors associated with mortality in a model that includes all the independent variables simultaneously, a multivariate analysis was conducted using a binary logistic regression. Variables that were significant in the bivariate analysis included the serum albumin concentration, blood urea nitrogen (BUN) concentration, L/A ratio, duration of mechanical ventilation (in days), vasopressor use and duration (in days), norepinephrine use, vasopressin use, steroid use, and mortality risk as calculated by the qSOFA score. The statistically significant results (*p* < 0.05) from the multivariate analysis are presented in Table 3.

## 4. Discussion

In our study, no significant differences were found between variables related to clinical history between the survivor and non-survivor groups. Lichtenauer et al. [27] found substantial age differences between survivors and non-survivors (63.67 vs. 67.67 years, *p* = 0.01). Our study’s sample size and the age of our hospital’s patient population may explain this disparity.

Survivors and non-survivors, in our study, had significantly different serum albumin (2.9 vs. 2.7 mg/L, *p* = 0.015) and blood urea nitrogen (BUN) values (30.7 vs. 36.8 mg/dL, *p* = 0.040). Statistically significant differences persisted when defining abnormal albumin levels as <3 mg/L (Table 2). This is consistent with prior studies linking low albumin levels to more severe septic disease [27,39] and with the findings reported in other studies, in which non-survivors have higher BUN values, indicating signs of decreased renal function [27]. BUN elevations often suggest renal impairment in sepsis and septic shock [6], and renal injury may cause inflammation-induced proteinuria and hypoalbuminemia [31]. Low serum albumin levels increase vascular endothelium permeability, exacerbating fluid buildup and decreasing circulatory function in sepsis [31]. This emphasizes the importance of monitoring critically ill patients’ albumin and renal function.

Our study found no statistically significant differences in serum lactate levels or with the percentage of patients with lactate levels >2 mmol/L, but there was a significant difference in the L/A ratio (0.60 vs. 0.95, *p* = 0.003); this is in contrast to other studies that found a significant association between lactate concentrations and sepsis-related mortality [27,38,39]. Steroid use, mechanical ventilation (MV) and its duration, and vasopressor (norepinephrine and vasopressin separately) use and duration were other clinical characteristics that differed significatively between survivors and non-survivors in our study, indicating a predictively worse clinical condition in non-surviving patients. In relation to the qSOFA, 74.2% of non-survivors were classified as high risk compared to 40.4% of survivors (*p* < 0.0001). Vasopressor use, the need for mechanical ventilation, and qSOFA score may reflect baseline patient severity rather than mortality predictors as it was not possible to separate patients with sepsis from patients with septic shock for our study.

Interestingly, while lactate levels did not show a statistically significant difference between groups, albumin levels did, with a greater and significant proportion of patients with hypoalbuminemia in the non-survivor group. However, after conducting an ROC curve analysis for each of these variables, only the L/A ratio demonstrated statistical significance in distinguishing between survivors and non-survivors.

The AUC of the L/A ratio in our study was 0.651 (95% CI: 0.55–0.74), consistent with previous research evaluating its prognostic value. For instance, Gharipour et al. reported that the L/A ratio provided a better discriminatory value for sepsis mortality than lactate alone (AUC: 0.63 vs. 0.60) [38]. Similarly, Chebl et al. (2021) found an AUC of 0.65 (95% CI: 0.61–0.70) for the L/A ratio, which was significantly higher than lactate alone (AUC: 0.60, 95% CI: 0.55–0.64, *p* < 0.001) [40].

Yucel et al. (2021) found a statistically significant increase in the L/A ratio in non-survivors compared to survivors, with an AUC of 0.690, supporting its high predictive potential in sepsis-related mortality [42]. Both the results of our study and prior research support the L/A ratio as a useful predictive biomarker [27,38]. Lactate, associated with metabolic acidosis [27,28], and albumin, associated with inflammation and vascular permeability [31], are combined in the L/A ratio and these processes are intimately linked to sepsis and septic shock pathophysiological alterations. This may explain their better prognostic power than lactate or albumin alone [27,38].

In our study, an L/A ratio cutoff of 1.12 distinguished survivors from non-survivors, with a sensitivity of 36% and a specificity of 82%. This value is similar to that reported by Gharipour et al. (a cutoff value of 1.01, sensitivity of 47%, and specificity of 78%) [38] and is slightly lower than those reported by Wang et al. (a cutoff value of 1.735, sensitivity of 100%, and specificity of 51%) [36], Lichtenauer et al. (cutoff value of 1.5 and significantly associated with a 54% mortality) [27], and Shin et al. (a cutoff value of 1.32, sensitivity of 66%, and specificity of 62%) [34].

The linear logistic regression showed no significant relation between mortality and the L/A ratio. This may suggest that clinical variables not accounted for in our study might have affected the L/A ratio.

Compared to the other significant variables in the bivariate logistic regression, albumin (a negative relationship) and vasopressor use (a positive relationship) have a greater impact on mortality, as shown in the multivariate analysis. This underlines the importance of albumin in the L/A ratio calculation and as an individual prognostic factor. The statistical significance of vasopressor use in non-surviving patients can be related to the more severe clinical condition of these patients, as mentioned above.

### Limitations and Future Directions

Our study has several limitations. Due to its retroactive design, it is not possible to rule out selection bias. Also, the population characteristics as well as the methodology for clinical laboratory sample collection, storage, and analysis could not be fully standardized. Furthermore, limited medical record data made it difficult to distinguish sepsis from septic shock patients. We could not consistently determine the sample collection time relative to the patient’s admission or the start of sepsis, making it impossible to compare the progression of patients’ illnesses. Finally, we did not account for total fluid administration or complications that may have affected clinical outcomes.

A larger-scale prospective study with standardized protocols is needed to further investigate the prognostic value of the L/A ratio [51]. These limitations should be carefully considered when interpreting our results.

Our study contributes to the evidence in favor of the usefulness of the L/A ratio as a prognostic indicator of mortality in sepsis, easily applicable even in resource-limited settings. Mexico’s national health spending has increased in recent years, but it remains below the Latin American and Caribbean average and far below the 2015 (OECD) average [23]. Public spending is 58% of health funding, while private contributions are mostly out-of-pocket; this contributes to the limited resources of public hospitals and organizations and consequently affects patient care and hinders resources for clinical investigations [23,52]. Therefore, despite their limitations, small-scale studies with robust methodologies are valuable in countries with limited research funding.

## 5. Conclusions

The L/A ratio was shown to be a useful discriminatory tool for predicting mortality in sepsis patients, with a cutoff value of 1.12, a sensitivity of 36.5%, and a specificity of 82%. Additionally, it has a better discriminatory performance in distinguishing survivors from non-survivors compared to albumin or lactate levels alone. Although the AUC (0.651) of the L/A ratio in our study indicates a low to moderate discriminatory capacity on its own, future research could provide further justification for its use for the prognostic evaluation of patients with sepsis in addition to the recommended prognostic scores, especially in resource-limited settings. These results should be confirmed through larger-scale prospective studies to further assess its reliability and applicability in diverse clinical settings.

## Figures and Tables

**Figure 1 jcm-14-02825-f001:**
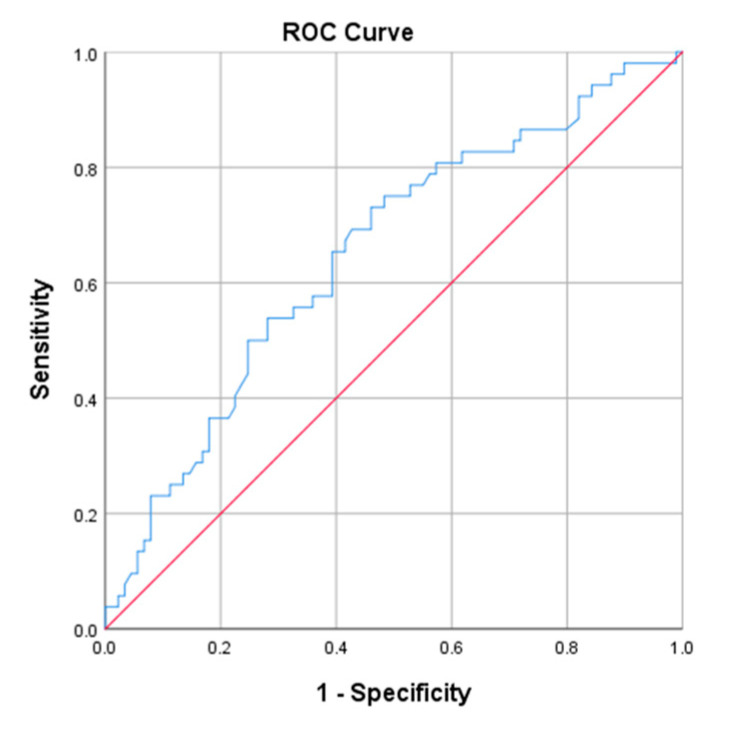
Receiver operating characteristic (ROC) curve for the L/A ratio in patients with sepsis.

**Table 1 jcm-14-02825-t001:** Comparison of the clinical and demographic characteristics of patients with sepsis between the survivor and non-survivor groups.

		Survivors (*n* = 89)		Non-Survivors (*n* = 52)		OR (CI 95%)	*p*
		Median	IQR	Median	IQR		
Age (years)		48	38.5, 60	53	42.25, 60.75	-	0.148
Weight (kg)		70	52.55, 84.50	62	54.25, 75	-	0.311
Height (m)		1.65	1.58, 1.7	1.66	1.6, 1.73	-	0.071
BMI (kg/m^2^)		24.81	20.76, 29.9	22.85	19.44, 26.88	-	0.117
		No.	%	No.	%		
Gender	Male	47	52.8	33	63.5	0.644 (0.320–1.299)	0.218
	Female	42	47.2	19	36.5		
Tobacco use	Yes	30	33.7	13	25	1.525 (0.709–3.282)	0.279
	No	59	66.3	39	75		
Alcoholism	Yes	25	28.1	12	23.1	1.302 (0.589–2.879)	0.514
	No	64	71.9	40	76.9		
Drug use	Yes	18	20.2	7	13.5	1.630 (0.631–4.212)	0.310
	No	71	79.8	45	86.5		
Comorbidities	Yes	72	81.8	41	78.8	1.207 (0.512–2.848)	0.667
	No	16	51.4	11	21.2		
Diabetes mellitus	Yes	35	39.3	25	48.1	1.429 (0.716–2.850)	0.311
	No	54	60.7	27	51.9		
Stroke	Yes	2	2.2	1	1.9	0.853 (0.075–9.642)	0.898
	No	87	97.8	51	98.1		
Systemic arterial hypertension	Yes	20	22.5	18	34.6	1.826 (0.856–3.897)	0.117
	No	69	77.5	34	65.4		
Site of infection	Lung	48	53.9	35	67.3	0.569 (0.279–1.161)	0.119
	Other	41	46.1	17	32.7		

IQR = interquartile range, CI = confidence interval, OR = odds ratio, BMI = body mass index.

**Table 2 jcm-14-02825-t002:** Comparison of the clinical and laboratory characteristics of patients with sepsis between the survivor and non-survivor groups.

		Survivors (*n* = 89)		Non-survivors (*n* = 52)		OR (CI 95%)	*p*
		Median	IQR	Median	IQR		
Albumin (mg/L)		2.90	2.45, 3.45	2.70	2, 3.07	-	0.015
INR		1.20	1.05, 1.34	1.29	1.16, 1.5	-	0.066
Glucose (mg/dL)		135	91–300	125	87, 203	-	0.222
Creatinine (mg/dL)		1.30	0.84, 2.13	1.40	0.84, 3.81	-	0.318
Urea (mg/dL)		63	37, 109	77	44.75, 130.75	-	0.063
BUN (mg/dL)		30.70	18.15, 54.44	36.8	24.25, 60.7	-	0.040
Hemoglobin (g/dL)		12.70	10.35, 14.6	12.60	10.02, 14.38	-	0.580
Platelets (×10^9^/L)		218	141, 312.5	187.40	117.5, 240.75	-	0.054
Leukocytes (×10^9^/L)		15.80	10.06, 20.59	13.31	9.36, 21.66	-	0.577
Lactate (mmol/L)		1.80	1.3, 2.7	2	1.4, 3.35	-	0.107
Bicarbonate (mmol/L)		22.40	17.5, 26.5	22	18.27, 25.87	-	0.924
L/A		0.60	0.43, 0.97	0.95	0.59, 1.33	-	0.003
Procalcitonine (μg/L)		1.29	0.17, 13.48	0.89	0, 6.98	-	0.157
MV duration (days)		0	0, 3	2	0, 9.75	-	<0.0001
Duration of vasopressor use (days)		0	0, 1	3	0, 5	-	<0.0001
		No.	%	No.	%		
MV	Yes	23	25.8	31	59.6	0.236 (0.144–0.490)	<0.0001
	No	66	74.2	21	40.4		
Vasopressor use	Yes	23	26.1	35	67.3	0.172 (0.081–0.364)	<0.0001
	No	65	73.9	17	32.7		
Norepinefrine use	Yes	23	25.8	35	67.3	5.908 (2.794–12.494)	<0.0001
	No	66	74.2	17	32.7		
Vasopresin use	Yes	16	18	26	50	4.563 (2.119–9.824)	<0.0001
	No	73	82	26	50		
Steroid use	Yes	12	13.5	24	46.2	5.5 (2.430–12.449)	<0.0001
	No	77	86.5	28	53.8		
Albumin cutoff	≤3	49	55.1	38	73.1	0.451 (0.215–0.947)	0.034
	>3	40	44.9	14	26.9		
Lactate cutoff	<2	48	53.9	20	38.5	1.873 (0.933–3.761)	0.076
	≥2	41	46.1	32	61.5		
qSOFA mortality risk	High	66	74.2	21	40.4	4.236 (2.043–8.785)	<0.0001
	Low	23	25.8	31	59.6		

IQR = interquartile range, CI = confidence interval, OR = odds ratio, BMI = body mass index, INR = international normalization ratio, BUN = blood urea nitrogen, MV = mechanical ventilation, L/A = lactate/albumin ratio, qSOFA = quick score for sepsis.

**Table 3 jcm-14-02825-t003:** Statistically significant results from the multivariate analysis of factors associated with mortality in sepsis patients.

	OR (CI 95%)	*p*
Albumin	0.559 (0.340–0.919)	0.022
Vasopressor use	5.750 (2.668–12.392)	<0.0001

OR = odds ratio, CI = confidence interval.

## Data Availability

The data presented in this study are available on request from the corresponding author due to privacy reasons.

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
