# Peer review of "Prognostic Value of the Lactate/Albumin Ratio in Sepsis-Related Mortality: An Exploratory Study in a Tertiary Care Center with Limited Resources in Western Mexico"

_jcm, 2025, doi:10.3390/jcm14082825_

Round 1

Reviewer 1 Report

Comments and Suggestions for Authors

This is a well-developed manuscript. I have some suggestions:

  1. In the abstract, the authors describe sepsis as a condition with high mortality worldwide that is not referred to in the introduction. Authors may define shortly what sepsis is, including some data regarding prevalence and mortality and especially data regarding Mexico (if studies have been developed).
  2. In the 1st paragraph of introduction, a citation should be added on the end of each sentence e.g. when authors describe the importance of early sepsis identification a study that supports that should be added.
  3. In line 55 authors describe that several biomarkers have been developed for the early diagnosis of sepsis, but only two references are cited. Authors should add the citation of each score developed regarding early sepsis diagnosis.
  4. A score that is widely used and is suggested by guidelines is the NEWS score. NEWS is an aggregate scoring system derived from six physiologic parameters and is preferred among most physicians. In a retrospective review of several scores in ED patients with sepsis, NEWS was the most accurate predictor (area under the receiver operating characteristic [AUROC] curve 0.904, 95% CI 0.805-0.913). Authors should add this score to their introduction alongside with the other scores. These are two citations that you can add:
    1. Evans L, Rhodes A, Alhazzani W, Antonelli M, Coopersmith CM, French C, Machado FR, Mcintyre L, Ostermann M, Prescott HC, Schorr C, Simpson S, Wiersinga WJ, Alshamsi F, Angus DC, Arabi Y, Azevedo L, Beale R, Beilman G, Belley-Cote E, Burry L, Cecconi M, Centofanti J, Coz Yataco A, De Waele J, Dellinger RP, Doi K, Du B, Estenssoro E, Ferrer R, Gomersall C, Hodgson C, Hylander Møller M, Iwashyna T, Jacob S, Kleinpell R, Klompas M, Koh Y, Kumar A, Kwizera A, Lobo S, Masur H, McGloughlin S, Mehta S, Mehta Y, Mer M, Nunnally M, Oczkowski S, Osborn T, Papathanassoglou E, Perner A, Puskarich M, Roberts J, Schweickert W, Seckel M, Sevransky J, Sprung CL, Welte T, Zimmerman J, Levy M. Surviving Sepsis Campaign: International Guidelines for Management of Sepsis and Septic Shock 2021. Crit Care Med. 2021 Nov 1;49(11):e1063-e1143. doi: 10.1097/CCM.0000000000005337. PMID: 34605781.
    2. Covino M, Sandroni C, Della Polla D, De Matteis G, Piccioni A, De Vita A, Russo A, Salini S, Carbone L, Petrucci M, Pennisi M, Gasbarrini A, Franceschi F. Predicting ICU admission and death in the Emergency Department: A comparison of six early warning scores. Resuscitation. 2023 Sep;190:109876. doi: 10.1016/j.resuscitation.2023.109876. Epub 2023 Jun 17. PMID: 37331563.
  5. Not only C-reactive protein (CRP), albumin, and blood lactate levels have been identified as independent prognostic indicators but Plasma procalcitonin and Mid-regional pro-adrenomedullin (MR-proADM) have been widely used but are not diagnostic for sepsis either. Nevertheless, the authors should add these two biomarkers to the manuscript.
  6. Regarding the clinical significance of the lactate levels except citation, 10 authors should add a metanalysis or a large clinical trial to support this data. Two citations that can be added are the two following, but authors can search for more up-to-date information:
    1. Casserly B, Phillips GS, Schorr C, Dellinger RP, Townsend SR, Osborn TM, Reinhart K, Selvakumar N, Levy MM. Lactate measurements in sepsis-induced tissue hypoperfusion: results from the Surviving Sepsis Campaign database. Crit Care Med. 2015 Mar;43(3):567-73. doi: 10.1097/CCM.0000000000000742. PMID: 25479113.
    2. Evans L, Rhodes A, Alhazzani W, Antonelli M, Coopersmith CM, French C, Machado FR, Mcintyre L, Ostermann M, Prescott HC, Schorr C, Simpson S, Wiersinga WJ, Alshamsi F, Angus DC, Arabi Y, Azevedo L, Beale R, Beilman G, Belley-Cote E, Burry L, Cecconi M, Centofanti J, Coz Yataco A, De Waele J, Dellinger RP, Doi K, Du B, Estenssoro E, Ferrer R, Gomersall C, Hodgson C, Hylander Møller M, Iwashyna T, Jacob S, Kleinpell R, Klompas M, Koh Y, Kumar A, Kwizera A, Lobo S, Masur H, McGloughlin S, Mehta S, Mehta Y, Mer M, Nunnally M, Oczkowski S, Osborn T, Papathanassoglou E, Perner A, Puskarich M, Roberts J, Schweickert W, Seckel M, Sevransky J, Sprung CL, Welte T, Zimmerman J, Levy M. Surviving Sepsis Campaign: International Guidelines for Management of Sepsis and Septic Shock 2021. Crit Care Med. 2021 Nov 1;49(11):e1063-e1143. doi: 10.1097/CCM.0000000000005337. PMID: 34605781.
  7. Common causes of increased lactate levels are: Cardiac arrest and Acute pulmonary edema, these two causes should be added in the manuscript in line 87.
  8. Regarding the prospective study by Yin et al. (line 103) authors should consider the number of patients the study includes and the values of sensitivity and specificity with the confidence interval values as presented in the study.
  9. In line 112 authors describe one of the earliest studies, but they cite three different studies (11,18,19). Authors should state which study is the earliest developed.
  10. Also, if this study is the 1st developed in Mexico/ Latin America, authors should note that in the objective of the manuscript.
  11. I strongly recommend that authors support the fact that the study was conducted in resource-limited settings (is there any published or other statistical data that supports this fact?). Additionally, I suggest that authors could add some details regarding the difficulties of conducting research in low-resource settings based on published data.
  12. Regarding patients with heart failure, pulmonary edema, and any other disease that is connected with increased lactate levels; were they included or not in the study? Authors should state that in the methods section.
  13. How was the diagnosis of diabetes mellitus and arterial hypertension defined: based on drug prescription and/or levels of Hba1c/ SBP and DBP? These definitions should be added to the manuscript.
  14. It is noted that All biochemical analyses were conducted following standard methodologies in the hospital's clinical laboratory (line 150). Authors should specify these methods.
  15. The qSOFA abbreviation is fully written more than once in the manuscript. Each abbreviation should be described once (where it is first presented) and then added to the only as an abbreviation. Moreover, a table with abbreviations is strongly recommended at the end of the manuscript.
  16. IQR should be presented as IQR: Q1- Q3 and not just as the value of IQR. Q1 is referred to 25% percentile and Q3 to the 75% percentile.
  17. Additionally, the number of participants is referred to in the method section in the abstract while in the manuscript in the results. It is strongly suggested to add the number of participants in the same section, e.g., results, both in the abstract and in the manuscript.
  18. The median age should be presented accompanied by the IQR value e.g. median age (Q1-Q3) and in the text.
  19. In the discussion authors should add the values of sensitivity and specificity of all published studies.

Author Response

We appreciate the time you've taken to share your valuable comments. We hope that the revised article, and the timely responses below, address the areas of improvement mentioned (all changes in the new version of the article can be seen using the "change control" in Word).

  1. In the abstract, the authors describe sepsis as a condition with high mortality worldwide that is not referred to in the introduction. Authors may define shortly what sepsis is, including some data regarding prevalence and mortality and especially data regarding Mexico (if studies have been developed).
    1. Response: In response to your comments, we have added the definition of sepsis at the beginning of the introduction, as well as epidemiologic data from the world and Mexico on the prevalence and mortality of sepsis.
  2. In the 1st paragraph of introduction, a citation should be added on the end of each sentence e.g. when authors describe the importance of early sepsis identification a study that supports that should be added.
    1. Response: Based on your recommendations, we have adjusted and added references where necessary to adequately support each statement.
  3. In line 55 authors describe that several biomarkers have been developed for the early diagnosis of sepsis, but only two references are cited. Authors should add the citation of each score developed regarding early sepsis diagnosis.
    1. Response: We have added the reference (and additional text that complements the idea in line 55) that serves as a reference for different biomarkers and their usefulness in sepsis. In the paragraph where we discuss prognostic scores, we have changed the citations to make it easier to find the reference for each score, and we have also added new text (also in response to your next comment) briefly describing Early Warning Scores (EWS).
  4. A score that is widely used and is suggested by guidelines is the NEWS score. NEWS is an aggregate scoring system derived from six physiologic parameters and is preferred among most physicians. In a retrospective review of several scores in ED patients with sepsis, NEWS was the most accurate predictor (area under the receiver operating characteristic [AUROC] curve 0.904, 95% CI 0.805-0.913). Authors should add this score to their introduction alongside with the other scores. These are two citations that you can add:
    1. Evans L, Rhodes A, Alhazzani W, Antonelli M, Coopersmith CM, French C, Machado FR, Mcintyre L, Ostermann M, Prescott HC, Schorr C, Simpson S, Wiersinga WJ, Alshamsi F, Angus DC, Arabi Y, Azevedo L, Beale R, Beilman G, Belley-Cote E, Burry L, Cecconi M, Centofanti J, Coz Yataco A, De Waele J, Dellinger RP, Doi K, Du B, Estenssoro E, Ferrer R, Gomersall C, Hodgson C, Hylander Møller M, Iwashyna T, Jacob S, Kleinpell R, Klompas M, Koh Y, Kumar A, Kwizera A, Lobo S, Masur H, McGloughlin S, Mehta S, Mehta Y, Mer M, Nunnally M, Oczkowski S, Osborn T, Papathanassoglou E, Perner A, Puskarich M, Roberts J, Schweickert W, Seckel M, Sevransky J, Sprung CL, Welte T, Zimmerman J, Levy M. Surviving Sepsis Campaign: International Guidelines for Management of Sepsis and Septic Shock 2021. Crit Care Med. 2021 Nov 1;49(11):e1063-e1143. doi: 10.1097/CCM.0000000000005337. PMID: 34605781.
    2. Covino M, Sandroni C, Della Polla D, De Matteis G, Piccioni A, De Vita A, Russo A, Salini S, Carbone L, Petrucci M, Pennisi M, Gasbarrini A, Franceschi F. Predicting ICU admission and death in the Emergency Department: A comparison of six early warning scores. Resuscitation. 2023 Sep;190:109876. doi: 10.1016/j.resuscitation.2023.109876. Epub 2023 Jun 17. PMID: 37331563.
    3. Response: Taking your recommendation into account, and as mentioned in our response to your previous comment, we have added new text describing the EWS, as well as the evidence presented by Covino et al. regarding the NEWS. We have already referenced the Evans et al. article in some sections of the article, and with respect to the NEWS, we believe it is more appropriate to cite the Covino et al. reference that you kindly provided, as well as the reference to the original article describing this score.
  5. Not only C-reactive protein (CRP), albumin, and blood lactate levels have been identified as independent prognostic indicators but Plasma procalcitonin and Mid-regional pro-adrenomedullin (MR-proADM) have been widely used but are not diagnostic for sepsis either. Nevertheless, the authors should add these two biomarkers to the manuscript.
    1. Response: Based on your recommendation, we have added other biomarkers, including procalcitonin and MR-proADM, as well as their respective references.
  6. Regarding the clinical significance of the lactate levels except citation, 10 authors should add a metanalysis or a large clinical trial to support this data. Two citations that can be added are the two following, but authors can search for more up-to-date information:
    1. Casserly B, Phillips GS, Schorr C, Dellinger RP, Townsend SR, Osborn TM, Reinhart K, Selvakumar N, Levy MM. Lactate measurements in sepsis-induced tissue hypoperfusion: results from the Surviving Sepsis Campaign database. Crit Care Med. 2015 Mar;43(3):567-73. doi: 10.1097/CCM.0000000000000742. PMID: 25479113.
    2. Evans L, Rhodes A, Alhazzani W, Antonelli M, Coopersmith CM, French C, Machado FR, Mcintyre L, Ostermann M, Prescott HC, Schorr C, Simpson S, Wiersinga WJ, Alshamsi F, Angus DC, Arabi Y, Azevedo L, Beale R, Beilman G, Belley-Cote E, Burry L, Cecconi M, Centofanti J, Coz Yataco A, De Waele J, Dellinger RP, Doi K, Du B, Estenssoro E, Ferrer R, Gomersall C, Hodgson C, Hylander Møller M, Iwashyna T, Jacob S, Kleinpell R, Klompas M, Koh Y, Kumar A, Kwizera A, Lobo S, Masur H, McGloughlin S, Mehta S, Mehta Y, Mer M, Nunnally M, Oczkowski S, Osborn T, Papathanassoglou E, Perner A, Puskarich M, Roberts J, Schweickert W, Seckel M, Sevransky J, Sprung CL, Welte T, Zimmerman J, Levy M. Surviving Sepsis Campaign: International Guidelines for Management of Sepsis and Septic Shock 2021. Crit Care Med. 2021 Nov 1;49(11):e1063-e1143. doi: 10.1097/CCM.0000000000005337. PMID: 34605781.
    3. Response: Following your guidance, we have added new references to support the use of lactate as a biomarker in sepsis (including that of Casserly et al. and citing that of Evans et al., which we had already cited in the article).
  7. Common causes of increased lactate levels are: Cardiac arrest and Acute pulmonary edema, these two causes should be added in the manuscript in line 87.
    1. Response: We have added what was suggested in the paragraph describing some of the causes of elevated serum lactate levels.
  8. Regarding the prospective study by Yin et al. (line 103) authors should consider the number of patients the study includes and the values of sensitivity and specificity with the confidence interval values as presented in the study.
    1. Response: Following your recommendations, we have added the requested information on the Yin et al. study.
  9. In line 112 authors describe one of the earliest studies, but they cite three different studies (11,18,19). Authors should state which study is the earliest developed.
    1. Response: We have corrected the reference to the above statement to specify the study to which we are referring.
  10. Also, if this study is the 1st developed in Mexico/ Latin America, authors should note that in the objective of the manuscript.
    1. Response: To our knowledge, only three other studies have been conducted on the prognostic value of the lactate-albumin ratio in sepsis in Mexican patients. Of these three studies, only one is indexed in PubMed. This information has been added to the Introduction, and the relevant results of these studies are included in the Discussion section.
  11. I strongly recommend that authors support the fact that the study was conducted in resource-limited settings (is there any published or other statistical data that supports this fact?). Additionally, I suggest that authors could add some details regarding the difficulties of conducting research in low-resource settings based on published data.
    1. Response: We have added the reference to support the statement about the low-resource setting in the Introduction. We also added a short paragraph in the limitations section briefly describing the difficulties associated with research in low- and middle-income countries and the difficulties of working in a low-resource setting such as Mexico.
  12. Regarding patients with heart failure, pulmonary edema, and any other disease that is connected with increased lactate levels; were they included or not in the study? Authors should state that in the methods section.
    1. Response: Following your recommendation, we have added a clarification to the Methods section stating that patients with any of the independent causes of elevated lactate were not actively excluded. However, none of the patients included had a prior diagnosis of heart failure, pulmonary edema, or other conditions associated with elevated lactate. Patients with diagnosed diabetes and a history of alcohol and drug use (known causes of elevated lactate) were included. However, there were no significant differences in any of these variables that could affect the results.
  13. How was the diagnosis of diabetes mellitus and arterial hypertension defined: based on drug prescription and/or levels of Hba1c/ SBP and DBP? These definitions should be added to the manuscript.
    1. Response: The diagnoses of diabetes mellitus and hypertension were based on information available in the electronic medical records; specifically, only a documented history of these conditions was recorded. As this was a retrospective study, detailed information regarding the diagnostic criteria used for each patient was not available. Consequently, the methodology section refers only to the collection of a "history of diabetes mellitus, arterial hypertension, stroke, and other comorbidities."
  14. It is noted that All biochemical analyses were conducted following standard methodologies in the hospital's clinical laboratory (line 150). Authors should specify these methods.
    1. Response: As with the documentation of known diagnoses, clinical laboratory data in this retrospective study were obtained directly from the electronic medical records. Consequently, we can only partially confirm that the tests were performed according to standard hospital laboratory protocols. Detailed descriptions of laboratory methodology such as sample collection, storage, and analysis are typically included in prospective studies, which can standardize and control these conditions. In contrast, in retrospective studies like ours, it is not possible to verify whether the same procedures were consistently applied for each patient. This represents an inherent limitation of retrospective study designs, as such methodological details are generally not documented in medical records. A brief statement acknowledging this limitation has been added to the limitations section.
  15. The qSOFA abbreviation is fully written more than once in the manuscript. Each abbreviation should be described once (where it is first presented) and then added to the only as an abbreviation. Moreover, a table with abbreviations is strongly recommended at the end of the manuscript.
    1. Response: We have corrected the unnecessary mention of the full abbreviation for qSOFA. We feel that for an original article of our length, it is unnecessary to include a table of abbreviations (it is not typically used in similar articles). If you still feel it is necessary, we could include it as an appendix to the article.
  16. IQR should be presented as IQR: Q1- Q3 and not just as the value of IQR. Q1 is referred to 25% percentile and Q3 to the 75% percentile.
    1. Response: Based on your suggestion and in accordance with APA style recommendations, we have corrected the IQR description.
  17. Additionally, the number of participants is referred to in the method section in the abstract while in the manuscript in the results. It is strongly suggested to add the number of participants in the same section, e.g., results, both in the abstract and in the manuscript.
    1. Response: We have removed the mention of the number of patients included in the methods section of the abstract, and only mention it in the results section of the abstract and in the full text.
  18. The median age should be presented accompanied by the IQR value e.g. median age (Q1-Q3) and in the text.
    1. Response: Based on your suggestion and in accordance with APA style recommendations, we have corrected the IQR description.
  19. In the discussion authors should add the values of sensitivity and specificity of all published studies.
    1. Response: We have added the sensitivity and specificity values (in studies where available) reported in the studies cited in the discussion (in relation to the L/A ratio cut-off).

Reviewer 2 Report

Comments and Suggestions for Authors

The chosen topic is of great interest in the field of critical patient care in the intensive care unit. The manuscript is well-written and structured, presenting the chosen topic in detail; however, it can be improved.

The manuscript introduction is well-structured and informative, but there are areas that could be improved for clarity, coherence, and readability, like the phrase "The prognosis of patients with sepsis can be significantly improved through early identification and management within the first few hours."

The sentence "Nonetheless, cost-effective biomarkers with adequate sensitivity and specificity for detecting high-risk patients are still being sought [1,2]." is vague, it can be rewritten.

The introduction transitions from sepsis prognosis to scoring systems but lacks a clear link between these elements and the lactate/albumin (L/A) ratio (which is the study’s focus).Adding a bridging sentence, introducing why alternative biomarkers, such as the L/A ratio, are worth exploring.

The text discusses existing scores in detail but does not explicitly state their limitations in a way that justifies investigating new biomarkers, something that I believe should be mentioned.

Some sentences are lengthy or contain redundant phrasing. Consider simplifying for readability,

  • like:  "Prognostic mortality indices are essential for recognizing patients requiring intensive clinical management and for optimizing the allocation of healthcare resources to prevent adverse outcomes." 
  • "While the correlation between lactate levels and mortality has been well established [2], its reliability as an isolated predictor of mortality remains controversial due to multiple confounding factors affecting lactate concentrations."

The transition from lactate to albumin is somewhat abrupt. Consider a linking sentence.

You mention that the L/A ratio has been studied in pediatric and adult populations but do not specify what gaps still exist.

The exclusion of patients with chronic kidney disease, nephrotic syndrome, liver failure, or chronic malnutrition is justified since these conditions could confound lactate/albumin ratios. However, it would be helpful to explain if any other potential confounders were considered for exclusion, particularly those that may impact lactate or albumin levels but are not included in the exclusion criteria.

The use of bivariate analysis (e.g., chi-square, Mann-Whitney U) is appropriate for initial comparisons. However, it's worth mentioning why a multivariate regression analysis was performed after this step. Were interactions between clinical variables and biomarkers (lactate, albumin) considered? I belive this could be emphasized to clarify the rationale for using multivariate logistic regression.

It’s mentioned that the cutoff value of the L/A ratio was determined based on sensitivity and specificity; however, the rationale for selecting a specific cutoff point could be elaborated upon. Why was this particular cutoff used? Were other thresholds tested for improved sensitivity or specificity?

In the beginning of the discussion, you contrast your findings with those of Lichtenauer et al., who reported significant differences in patient age. While the comparison is good, it might be helpful to discuss potential reasons why your study did not find a significant age difference. Perhaps the patient population or other demographic variables in your study differ. Emphasizing these differences can provide valuable context to your findings and enhance the strength of your interpretation.

Refine the interpretation of albumin and BUN: When discussing the significant differences in albumin and BUN levels, it might be useful to briefly explain why these variables are clinically important. For instance:

  • Albumin: Low serum albumin levels are associated with increased permeability of the vascular endothelium, which can exacerbate fluid accumulation and further compromise the circulatory system in sepsis. This reinforces the importance of monitoring albumin levels in critically ill patients.
  • BUN: Elevated BUN levels often reflect renal dysfunction, which is common in sepsis and septic shock. It may be worth noting how kidney function impacts overall outcomes in sepsis

Discuss the L/A ratio more explicitly. While you mention the L/A ratio's significance, adding a sentence to explain why it could be a better predictor of mortality in sepsis might be helpful. The L/A ratio combines two markers that are associated with different physiological processes (lactate with metabolic acidosis and albumin with inflammation and vascular permeability). This combination may explain why it has a stronger predictive ability than lactate alone.

This refined discussion strengthens the connections between your findings, clinical implications, and previous research while acknowledging the study's limitations. It also enhances the clarity of the interpretation of results and their potential application

The conclusions written based on your research can be improved and made clearer.

For example:

  • Sensitivity and Specificity Clarification. You mention that the L/A ratio demonstrated better discriminatory performance compared to albumin or lactate levels alone. While this is a good point, it might be useful to briefly mention why the sensitivity (36.5%) is relatively low. For example, you could add a sentence explaining that the lower sensitivity is expected in some prognostic tests, but the high specificity (82%) suggests its value in confirming mortality when a high risk is indicated.
  • While you mention the potential clinical utility of the L/A ratio, you could strengthen the conclusion by highlighting how your findings could influence current clinical practices or decision-making in sepsis management.

Overall the conclusion is well written and does a good job summarizing your study's findings while pointing out the need for further research. With the minor suggestions, the conclusion could become even more compelling by providing a clearer picture of the study's impact and how the findings might influence future sepsis management or clinical practice. Overall, great work!

Author Response

We appreciate the time you've taken to share your valuable comments. We hope that the revised article, and the timely responses below, address the areas of improvement mentioned (all changes in the new version of the article can be seen using the "change control" in Word).

  1. The manuscript introduction is well-structured and informative, but there are areas that could be improved for clarity, coherence, and readability, like the phrase "The prognosis of patients with sepsis can be significantly improved through early identification and management within the first few hours."
    1. Response: Following your recommendation, we have amended this and a number of other statements to improve their clarity, readability and consistency.
  2. The sentence "Nonetheless, cost-effective biomarkers with adequate sensitivity and specificity for detecting high-risk patients are still being sought [1,2]." is vague, it can be rewritten.
    1. Response: Following your recommendation, we have amended this and a number of other statements to improve their clarity, readability and consistency.
  3. The introduction transitions from sepsis prognosis to scoring systems but lacks a clear link between these elements and the lactate/albumin (L/A) ratio (which is the study’s focus).Adding a bridging sentence, introducing why alternative biomarkers, such as the L/A ratio, are worth exploring.
    1. Response: We have added a better link between the above scores and biomarkers in the Introduction section, as well as a rationale for the importance of looking for biomarkers such as the L/A ratio.
  4. The text discusses existing scores in detail but does not explicitly state their limitations in a way that justifies investigating new biomarkers, something that I believe should be mentioned.
    1. Response: We've added a rationale for the importance of looking for biomarkers such as the L/A ratio. We have modified the paragraph to better address the general limitations of prognostic scores.
  5. Some sentences are lengthy or contain redundant phrasing. Consider simplifying for readability,
    1. like:  "Prognostic mortality indices are essential for recognizing patients requiring intensive clinical management and for optimizing the allocation of healthcare resources to prevent adverse outcomes." 
    2. "While the correlation between lactate levels and mortality has been well established [2], its reliability as an isolated predictor of mortality remains controversial due to multiple confounding factors affecting lactate concentrations."
    3. Response: Following your suggestion, we have improved the flow of the text, added linking clauses, corrected some sentences (including the ones you mention) and added new paragraphs to help create a smoother transition within the text.
  6. The transition from lactate to albumin is somewhat abrupt. Consider a linking sentence.
    1. Response: We have placed a statement to better link these two issues.
  7. You mention that the L/A ratio has been studied in pediatric and adult populations but do not specify what gaps still exist.
    1. Response: We have added a short statement indicating some of the gaps in the population in which the L/A ratio has been studied.
  8. The exclusion of patients with chronic kidney disease, nephrotic syndrome, liver failure, or chronic malnutrition is justified since these conditions could confound lactate/albumin ratios. However, it would be helpful to explain if any other potential confounders were considered for exclusion, particularly those that may impact lactate or albumin levels but are not included in the exclusion criteria.
    1. Response: Following your recommendation (which coincides with what another reviewer mentioned), we have added to the Methods section some of the diseases that could affect albumin and lactate levels, which were not considered exclusion criteria, but were not found in the clinical records of the patients included in the study.
  9. The use of bivariate analysis (e.g., chi-square, Mann-Whitney U) is appropriate for initial comparisons. However, it's worth mentioning why a multivariate regression analysis was performed after this step. Were interactions between clinical variables and biomarkers (lactate, albumin) considered? I belive this could be emphasized to clarify the rationale for using multivariate logistic regression.
    1. Response: Multivariate binary logistic regression analysis was performed using a backward stepwise likelihood ratio method (described in the methods) because of the exploratory nature of the relationship between the independent variables and the binary dependent variable (mortality). This method has the advantage of being an iterative method, performed in steps, starting with the inclusion of all variables in the model (all independent variables included) and testing the significance of the inclusion of each variable in the full model by means of the likelihood ratio test. The variable with the least significant contribution (highest p-value) is removed from the model and a new step is performed with the remaining variables. This process is repeated until no variables can be eliminated, i.e. until only variables with a significant p-value remain. The final model contains only those variables that contribute significantly to a predicted outcome. The advantage of this method is that it provides a very parsimonious and efficient model, verified by the Hosmer-Lemeshow test. Therefore, it provides different information than bivariate analysis, as it tests all variables simultaneously and thus provides the variables that contribute most significantly to the outcome (mortality). We have added a short statement in the results section to remind people of what was described in the methodology and to streamline the presentation of the results. We have also expanded the interpretation of the bivariate logistic regression results in the Discussion section to improve clarity.
  10. It’s mentioned that the cutoff value of the L/A ratio was determined based on sensitivity and specificity; however, the rationale for selecting a specific cutoff point could be elaborated upon. Why was this particular cutoff used? Were other thresholds tested for improved sensitivity or specificity?
    1. Response: Youden's J statistic was used to select the ROC curve cut-off point with the best balance of sensitivity and specificity. We have included this specification in the Methods section.
  11. In the beginning of the discussion, you contrast your findings with those of Lichtenauer et al., who reported significant differences in patient age. While the comparison is good, it might be helpful to discuss potential reasons why your study did not find a significant age difference. Perhaps the patient population or other demographic variables in your study differ. Emphasizing these differences can provide valuable context to your findings and enhance the strength of your interpretation.
    1. Response: We have added a possible explanation for the differences in age found in our study.
  12. Refine the interpretation of albumin and BUN: When discussing the significant differences in albumin and BUN levels, it might be useful to briefly explain why these variables are clinically important. For instance:
    1. Albumin: Low serum albumin levels are associated with increased permeability of the vascular endothelium, which can exacerbate fluid accumulation and further compromise the circulatory system in sepsis. This reinforces the importance of monitoring albumin levels in critically ill patients.
    2. BUN: Elevated BUN levels often reflect renal dysfunction, which is common in sepsis and septic shock. It may be worth noting how kidney function impacts overall outcomes in sepsis
    3. Response: We have clarified the interpretation of albumin and BUN results in the discussion following your recommendations.
  13. Discuss the L/A ratio more explicitly. While you mention the L/A ratio's significance, adding a sentence to explain why it could be a better predictor of mortality in sepsis might be helpful. The L/A ratio combines two markers that are associated with different physiological processes (lactate with metabolic acidosis and albumin with inflammation and vascular permeability). This combination may explain why it has a stronger predictive ability than lactate alone. This refined discussion strengthens the connections between your findings, clinical implications, and previous research while acknowledging the study's limitations. It also enhances the clarity of the interpretation of results and their potential application
    1. Response: Following your valuable suggestion, we have added this explanation of a possible cause for the prognostic superiority of the L/A ratio (with the appropriate citations to support it).
  14. The conclusions written based on your research can be improved and made clearer. For example:
    1. Sensitivity and Specificity Clarification. You mention that the L/A ratio demonstrated better discriminatory performance compared to albumin or lactate levels alone. While this is a good point, it might be useful to briefly mention why the sensitivity (36.5%) is relatively low. For example, you could add a sentence explaining that the lower sensitivity is expected in some prognostic tests, but the high specificity (82%) suggests its value in confirming mortality when a high risk is indicated.
    2. While you mention the potential clinical utility of the L/A ratio, you could strengthen the conclusion by highlighting how your findings could influence current clinical practices or decision-making in sepsis management.
    3. Response: In order to strengthen the text in the concluding section, we have carefully followed your valuable suggestions.

Round 2

Reviewer 1 Report

Comments and Suggestions for Authors

Dear authors,
Most of my issues have been addressed. I have only one suggestion: you could add a graphical abstract to increase the reader's interest in your research article. I genuinely believe that this would be very helpful. 

Author Response

Dear authors,
Most of my issues have been addressed. I have only one suggestion: you could add a graphical abstract to increase the reader's interest in your research article. I genuinely believe that this would be very helpful. 

Response: We appreciate the suggestion. In our response, we have included the graphic abstract of the article that meets the editorial requirements (Is in pdf format just for the response. We submited a PNG format with all requirements with the resubmission).
